# COVID-19-Associated Encephalopathy (COVEP): Basic Aspects of Neuropathology

**George S. Stoyanov** [1,*], **Dimo Stoyanov** [2], **Martin Ivanov** [2], **Anton B. Tonchev** [2], **Hristo Popov** [1] **and Lilyana Petkova** [1]

[1] Department of General and Clinical Pathology, Forensic Medicine and Deontology, Faculty of Medicine, Medical University—Varna "Prof. Dr. Paraskev Stoyanov", 9002 Varna, Bulgaria

[2] Department of Anatomy and Cell Biology, Faculty of Medicine, Medical University—Varna "Prof. Dr. Paraskev Stoyanov", 9002 Varna, Bulgaria

[*] Correspondence: georgi.stoyanov@mu-varna.bg

**Definition:** SARS-CoV-2, a member of the betacoronavirus group and causative agent of COVID-19, is a virus affecting multiple systems, not only the respiratory. One of the systems affected by the virus is the central nervous system, with neuropathological studies reporting a wide set of morphological phenomena—neuroinflammation, vascular and blood-brain barrier alterations, neurodegeneration, and accelerated aging, while contradicting data is present on the direct neuroinvasive potential of the virus and active viral replication within neurons. The depicted changes, other than an acute effect (which may contribute to the death of the patient) also have chronic sequelae in the context of post-COVID syndrome cognitive impediments, sleep, and mood disorders. The following chapter describe the basic neuropathological aspects of SARS-CoV-2 as based on the present evidence in scientific literature and propose the term COVEP—COVID-associated encephalopathy—to unite the undisputed effects of the infection on nervous system morphology and function.

**Keywords:** SARS-CoV-2; COVID-19; central nervous system; anosmia; neuropathology; encephalopathy

## 1. Introduction

The novel coronavirus disease, which began its outbreak in 2019 (COVID-19) caused by severe acute respiratory syndrome coronavirus 2 (SARS-CoV-2), has emerged as a multisystem disease [1–3]. While initially regarded as a respiratory system disease, one of the earliest peculiarities of the disease was the development of anosmia (loss of smell sensation) and ageusia (loss of taste perception) by those infected, which pointed toward nervous system involvement as well [4–9]. Since then, many other aspects of the disease have emerged, such as renal involvement (COVAN), myopericarditis and endotheliitis, gastrointestinal system disorders, and many more [10–17]. With the emergence of new variants of SARS-CoV-2 that cause severe disease in significantly fewer people while also being severely more infective, the scientific focus has gradually shifted toward the chronic sequelae of infection: post-COVID syndromes where once more the nervous system takes center stage [18–23].

## 2. Applications and Influences

As a multisystem infection, it is essential not to underestimate or focus the chronic or acute sequelae of COVID-19 purely to the respiratory system [21,23–25]. A myriad of symptoms in the acute and chronic phases of the disease show signs of its effect on the central nervous system—anosmia, ageusia, migraine headaches, confusion, clouding of consciousness, memory problems, etc., [8,22,26–28].

## 3. COVID-19 and the Brain

Coronaviruses, as a diffuse group of viruses, have shown many properties and organ targets in their biology [3,6,29–31]. The previous two coronaviruses of concern—severe acute respiratory syndrome coronavirus (SARS) and middle eastern respiratory syndrome coronavirus (MERS)—also had signs pointing toward central nervous system involvement, including morphological ones [26,29,32–34]. Furthermore, close relatives to SARS-CoV-2 have been known to cause encephalitis in other animals, such as the mouse hepatitis virus, which is known to have significant neurotropic properties [35–37].

The mechanisms of neuroinvasion to this point are still not well known, but there are several candidates, including direct invasion through the cribriform plate, neuronal pathway invasion, and hematogenous dissemination through COVID-19-induced endothe-liitis [38–40]. Direct invasion through the cribriform plate and nasal to meningeal dissemi-nation is a well-known mechanism utilized by viruses, including some other members of the coronavirus family and bacteria [22,38,40–42]. Direct neuronal invasion is a mechanism typical for neurotropic viruses, in which once the viral particles enter the neuronal body they travel antero- and retrograde through it, with the mechanism being suggested not only for the nasal cavity nerve endings and bodies but also for pulmonary and gastrointestinal ones as well; after this, the newly produced viral particles infect neighboring neurons and repeat the chain [22,43–46]. The third and most important mechanism, as it plays a role not only in the direct neuronal involvement of SARS-CoV-2 but also in its myriad of vascular complications, is that of endothelialitis, in which the endothelial cells of the blood ves-sels are infected and undergo subsequent desquamation and necrosis [22,29,38,39,47–49]. Neuroinvasion through this mechanism would induce direct viral damage to the central nervous system neurons and toxic harm as well, as the endothelial cell damage would disrupt the blood–brain barrier [50–54]. Limited data on the choroid plexus further suggests that there may be choroid tropism of the virus, with secondary blood-cerebrospinal fluid disruption [55].

Another suggested mechanism by which SARS-CoV-2 can affect the central ner-vous system is that of inflammatory cell mediation. Direct immune cell invasion, although primarily affecting the immune response and cytokine storm onset, is also an important mechanism by which unaffected organs can be directly infected by viral shedding from circulating infected immune cells, especially if they migrate to the parenchyma [56,57].

Further unknowns are raised by the exact mechanism by which SARS-CoV-2 can affect the functioning of the central nervous system. While neuroinvasion refers to the ability of the viral particles to enter the CNS, that by itself does not directly indicate a pathological sequence [58]. Neurotropism on the other hand reflects on the ability of the virus to successfully enter the CNS cells, while neurovirulence reflects on the pathogenic capabilities of the virus in the nervous system—direct or indirect—without neurotropism [58]. While undoubtably SARS CoV-2 has neuroinvasive potential through multiple mechanisms, the neuroinvasive aspects have been confirmed only in vitro, with most postmortem studies disputing this mechanism [43,59,60]. On the other hand, multiple postmortem studies indicate a myriad of pathological changes, ranging from necrotic and degenerative to inflammatory and reactive, indicating that even in the absence of neurotropism, there are signs of neurovirulence [38,58,61,62].

### 3.1. Meninges

Reports on meningitis and meningeal involvement in COVID-19 are relatively scarce. Most scientific literature reports signs of meningism—severe headaches, disorientation, and neck stiffness, with some also noting seizures [26,38,63]. In most of these cases, aseptic meningitis is diagnosed in a predominantly severe clinical course setting, although in some patients, meningism may be the initial presentation of the disease [26,63]. Only a handful of reports were able to isolate SARS-CoV-2 from the cerebrospinal fluid of such patients [64,65]. On the other hand, there are multiple reports of co-infection meningitis

occurring in sepsis and in secondary bacterial infections during COVID-19 [66]. Most morphological studies reporting cases without morphological evidence of a superimposed infection report meningeal histopathological changes consisting of severe edema, focal microhemorrhages, and lymphocytic infiltration [9]. Involvement of the meninges based on the endotheliitis mechanism is most likely to explain the clinical symptoms as based on the morphology—vascular damage would cause the edema and lymphocytic infiltration, with only those having sufficient hemorrhages, i.e., viral particles entering from the blood—giving a positive result upon laboratory test for SARS-CoV-2 from the cerebrospinal fluid [8,13,16,39,47]. Of course, the direct invasion through the cribriform plate is also a viable and highly likely mechanism, although it is ill-supported by the morphological findings [8].

### 3.2. Cerebrum

As in nearly all aspects of neuropathology, the gross changes in the brain are mostly minimal and predominantly nonspecific, manifested by cerebral edema resulting in cerebellar tonsillar herniation into the foramen magnum, edema of the meninges, and focal vascular changes, predominantly petechial hemorrhages in severe cases [67].

The focal point of COVID-19-induced neuropathological damage is, of course, the direct effects of the virus on the central nervous system, predominantly the cerebrum. These can be viewed in several different mechanisms; the first and most profound clinically is that of hypoxia-induced hypoperfusion due to the pulmonary changes induced by the virus; due to decreased oxygen saturation, neurons start undergoing hypoxic and hence degenerative change, with this mechanism being most prominent in patients with a severe clinical course—hypoxic brain injury [50–53]. The second mechanism corresponding well to the clinical manifestation is that of endothelialitis, where the desquamated endothelial cells cause a predisposition for thrombosis and vascular ruptures, resulting in the clinically well-documented higher incidence of cerebrovascular disease in COVID-19 [68–70]. This mechanism would also lead to blood–brain barrier disruption and the onset of neuronal degeneration through toxin-induced damage [22,53,71]. The third and final mechanism is the direct neuronal invasion of SARS-CoV-2 and viral-induced damage to neurons—viral encephalitis [22,29,38].

### 3.2.1. Cerebral Hypoxic Brain Injury and Hypoperfusion

Hypoxic brain injury, also referred to as anoxic and hypoxic-ischemic brain injury, is a complex neuropathological process in which neurons undergo first degenerative and, subsequently, necrotic change due to insufficient oxygen supplementation [72]. The causative agents for the decreased oxygen saturation or vascular perfusion are multiple and vary from direct reduction of plasma oxygen, as in COVID-19, temporary vascular obstruction or flow reduction, as in trauma, to cardiac arrest, carbon monoxide poisoning, etc., [50,71–73]. This mechanism would directly correlate with the morphology observed in fatal cases of COVID-19, as those patients typically have a significant decrease in oxygen saturation [74,75]. Interestingly, the long-term consequences of hypoxic brain injury correlate highly to the so-called "brain fog" in the context of post-COVID syndrome—transient clouding of consciousness, cognitive disabilities, attention deficit, sleep disorders, executive dysfunction, etc., [62,76]. These mechanisms of brain injury are well-supported by neuropathological data depicting foci of micronecrosis and hemorrhages, axonal degeneration and microgliopathy [54,62,77–79]. However, they would not apply to people experiencing the same symptoms in the context of post-COVID syndrome without suffering from a severe form of the disease in the acute phase, with significant pulmonary involvement and hypoxia [22,24,80].

### 3.2.2. Vascular Incidents

As mentioned several times, endotheliitis is a focal point of SARS-CoV-2-induced organ pathology, and the central nervous system is no exception here [13,16,39,47–49,70,81,82]. Other than disruption of the structure of the blood–brain barrier, direct viral replication within the vascular endothelial cells and their following desquamation and necrosis results in exposing the endothelial cell basal lamina collagen, which acts as a tissue factor in the activation of the coagulation cascade and hence enacts Virchow's triad, leading to vascular thrombosis formation [83–87]. As a result, the autochthonous thrombi in predominantly small and medium-sized blood vessels lead to focal ischemia and neuronal necrosis—cerebral infarction (clinically referred to as stroke), a well-reported sequence of SARS-CoV-2 infection [22,29,68–71]. Besides autochthonous thrombi, the multiple systems involved by endotheliitis and further cardiovascular sequelae also allow for more significant ischemic incidents due to thromboembolism to the brain [68,78,88]. Other than ischemic incidents, these patients also have a higher likelihood of developing cerebral hemorrhages (clinically referred to as hemorrhagic stroke), as well as thrombosis of the dural sinuses (cerebral venous sinus thrombosis) [68,71].

The extent of microvascular damage to the central nervous system should also not be underestimated, even when there is a lack of microvascular thrombi, as some capillaries in the context of endotheliitis completely lose their endothelium and collapse, resulting in a string-like transformation [89]. These may play a key role in COVID-induced neuropathological damage, as these inactive and disappearing capillaries induce hypoxia in the peripheral tissues [90].

### 3.2.3. Encephalitis and Encephalopathy

As already mentioned, a focal point of research is the direct neurotropic effect of SARS-CoV-2 and its potential to lead to encephalitis, as in some of its close relatives [29,35,36,91–93]. In the initial phases of the pandemic, multiple studies reported encephalitis in COVID-19 patients ranging from panencephalitis to acute hemorrhagic, acute necrotizing, limbic, and autoimmune encephalitis [82,91–102]. In the case of confirmed limbic and autoimmune encephalitis, SARS-CoV-2 can be both a triggering and a concomitant factor, as has been reported with other viruses [103,104]. For necrotizing and hemorrhagic-necrotic encephalitis cases, the relationship has since become more dubious in the absence of histology, as most of these reports are based on neuroradiology and can be mimicked by the systemic and vascular effects of the virus [82,97–99]. The relationships and capabilities of SARS-CoV-2 to lead to the development of true encephalitis are further disputed by most morphological studies, as despite histology the morphological changes may be highly similar to those of encephalitis, and while there are data on micro- and astroglyopathy, as well as cytotoxic T lymphocyte infiltration, immunologically only a few studies have isolated viral presence in the central nervous system [9,62,67,78,105,106]. Worthy of note is the heterogeneity of reported locations of encephalitis by some studies; while most depict the brainstem as the most severely affected location, others point toward the limbic systems or isolated cortical and subcortical areas of the frontal lobe [9,89,94,98,104,107]. Here it is essential to mention that while neuroinflammation and encephalitis are often used as synonyms and have similar mechanisms and morphology, encephalitis refers to active infection-induced inflammation of the brain parenchyma, while neuroinflammation (clinically referred to as encephalopathy in some cases, i.e., hepatic encephalopathy) refers purely to the presence of an inflammatory reaction in the brain, predominantly due to neuronal loss leading to the release of inflammatory cytokines, as seen in a myriad of conditions—neurovirulence [108–110]. Based on the presence of inflammatory reaction, lack of undisputable evidence of the viral presence, and the evident neuronophagia overpowering the inflammatory response, these data point more towards encephalopathy than encephalitis [111–116].

### 3.2.4. Neurodegeneration

Given the mechanisms described so far and their inevitable neuronal degeneration and necrosis spectrum of sequelae, they are predominantly the ones occurring in severe and fatal cases [26,68,89]. Accumulating evidence suggests additional mechanisms associated with SARS-CoV-2 infection that could trigger or accelerate already-developed neurodegenerative disorders [117,118].

Some studies suggest that the viral surface structure acts as a binding site for heparin and accelerates the aggregation and displacement of heparin-binding proteins such as Aβ, α-synuclein, tau, prion, and TDP-43 RRM, resulting in initiation or acceleration of the associated neurodegenerative pathways [119,120]. Other studies have also suggested that within the host cells, the SARS-CoV-2 viral protein interacts with multiple intracellular pathways associated with aging, taking part in vesicle trafficking, lipid modifications, RNA processing and regulation, ubiquitin ligases, and mitochondrial activity, while at the same time depleting intracellular iron and further hindering mitochondrial function [59,60,117]. Furthermore, as the virus enters the host cell through the ACE-2 receptor, viral interaction may lead to decreased functionality, reducing the neuroprotective effects of ACE-2 [121,122]. These mechanisms would have implications for the central nervous system and multiple other organ sites, where resident cellular mitotic activity is low. However, based on the dispute over viral presence within neurons, this intracellular pathway of accelerated aging is dubious at this point and heralds significant future research [78,123].

One group of patients with neurodegenerative diseases severely affected by SARS-CoV-2 infection are those with Parkinson's disease. Multiple case reports and cohort studies have reported a significant functional and cognitive decline in these patients, with these patients also being more likely to expire from COVID-19 [124,125]. The exact mechanism of accelerated degeneration in these patients is also highly disputed, with some studies supporting the direct neuroinvasive potential of SARS-CoV-2 and pointing toward the brainstem as the most affected area in the encephalitis spectrum; others propose an ACE-2 suppression-associated mechanism, as the receptor is significantly expressed in dopaminergic neurons, while others point to the fact that infection-associated acceleration in Parkinson's disease is a typical hallmark of the disease and that progressive decline in these patients is predominantly due to physical and other therapies being discontinued during lockdowns [117,124–128]. One further fact supporting these claims is the isolation of coronavirus antibodies against other members of the viral family from the cerebrospinal fluid of patients with Parkinson's disease in the past decades [128,129].

Furthermore, discontinuation of non-medication treatment and social isolation during lockdowns has been shown to have adverse functional effects, not only on patients with chronic diseases, including neurodegenerative ones such as Alzheimer's disease, but significantly on neuropsychiatric diseases such as depression, anxiety, and post-traumatic stress disorder [117].

### 3.2.5. Hippocampal Damage

The hippocampus is a small and fragile structure within the brain with vast implications for neurodegenerative and other conditions, as it is a crucial area for adult neurogenesis [130]. As the hippocampus has been identified to play a role in a myriad of conditions, some studies have also tried to elaborate on its possible damage in the context of SARS-CoV-2 infection, as some of the long-term sequelae of the infection, such as brain fog, are indicative of hippocampal damage [62,130,131]. Clinical studies have established hippocampal hyperintensity and atrophy on brain MRI scans, suggesting a de novo process during COVID-19 [132]. Morphological studies have shown aberrant permeability of the blood–brain barrier due to endotheliitis, perivascular and pericellular microcalcifications, and neuronal loss in the dentate gyrus resulting in "skip" lesions and loss of doublecortin-positive immature neurons [133,134]. Although no viral RNA was isolated,

the hippocampus in these studies showed microglial activation, T lymphocyte infiltration, and increases in the levels of interleukins 1 and 6, suggestive of a neuroinflammation cascade, with the result also being recreated in an animal model of SARS-CoV-2 infection—neurovirulence [133]. Furthermore, spatial and architectural changes were noted not only in the neuronal and microglial but also in the astrocytic cellular populations, as well as changes in the cellular morphology [135,136].

### 3.3. Olfactory Epithelium and Olfactory Bulbs

One of the most striking early features that first suggested nervous system involvement in COVID-19 was the onset of anosmia and ageusia, features rarely reported in the previous two coronaviruses of concern—SARS-CoV and MERS-CoV [7,8,27–29,137,138].

SARS-CoV-2, as a virus with undisputed epitheliotropic features like most members of the coronavirus family, was quickly established to infect respiratory epithelial cells, with the nasal cavity and pharynx epithelia being no exception; hence the nasal swab tests for diagnosis proved highly efficient [139–141].

While initially regarded as due to nasal obstruction, it soon became apparent that anosmia develops due to complex mechanisms [8,131]. Initial reports on olfactory epithelia showed olfactory mucosa involvement, with active viral replication in sustentacular cells (sensory neuron supporting cells), but not esthesiocytes (olfactory sensory neurons), with sustentacular cell involvement also allowing for probable leptomeningeal spread as well (Figure 1) [8,141]. As the sustentacular cells have many functions, among which are odorant endocytosis, local homeostasis regulation, and esthesiocyte cilia structural support, a functional aberration in them inhibits the proper function of the receptor neurons [8]. Further clinical-based neuroradiology reports described olfactory bulb edema and subsequent atrophy during and after infection [9,142–144]. Initial histopathology suggested changes associated with direct viral-induced damage to the bulbs with edema, inflammatory cell infiltration, microglial nodules and severe neuronal degeneration, and neuronal necrosis with areas of colliquation, suggestive of necrotizing olfactory bulbitis (Figure 1) [9]. Further reports supported the presence of neuroinflammation and degenerative changes within the olfactory bulbs and additional endotheliitis-associated blood–brain barrier injury and suggested the presence of viral RNA (Figure 1) [61,78,105,145–148]. A promising sign for the reversible nature of SARS-CoV-2 induced nervous system pathology is the recovery of smell sensation, albeit delayed in most patients after COVID-19, probably linked to adult neurogenesis in the bulb [7,149,150]. This mechanism would also be in play in the regeneration of the esthesiocytes and their sustentacular cell in the respiratory part of the nasal mucosa [151]. However, one of the most detailed studies to date indicated that there are no pathological sequelae in the olfactory nerves and bulb, but rather that all of these changes are attributed to the nasal mucosa infection [141].

Unlike anosmia, the mechanisms for ageusia remain widely unelaborated upon (Figure 1) [152–154].

## COVID-19-associated anosmia and ageusia

**Figure 1.** Mechanisms of anosmia and ageusia onset in COVID-19; red denotes disputed mechanisms.

### 3.4. Cerebellum, Spinal Cord and Peripheral Nervous System

Few studies have focused on the cerebellum, presenting predominantly neuroinflammatory change [67,105,155]. Interestingly enough, there have been clinical reports on the onset of cerebellar dysfunction leading to ataxia, with some reports also focusing on accelerated aging and neurodegeneration [101,156–158].

At this stage, severe morphological lesions induced in the spinal cord by SARS-CoV-2 infection have not been proven, with the most frequently reported vascular and ischemic incidents being meningeal and minor neuroinflammation with accompanying degenerative changes [77,89,107,159]. Individual cases of transverse myelitis have also been reported associated with the infection [160,161].

Peripheral neuropathy is a relatively common complication, especially in critically ill patients, and tends to correspond with the extent of time spent in intensive care units [161]. Direct damage to the peripheral nerves is also discussed, with some studies isolating viral particles and RNA from peripheral nerves, including cranial ones [159,161–163]. The resulting neuritis has been reported to affect the musculoskeletal system with the onset of type two muscle atrophy [163]. Further supporting the presence of peripheral nervous system involvement, direct or indirect, is the increased likelihood of COVID-19 patients developing Guillain-Barré syndrome [22,38,106,131].

### 3.5. Post-COVID Syndrome and Brain Fog

Post-COVID syndrome, also referred to as long COVID and persistent COVID, represents a wide group of symptoms and complaints reported by patients after acute SARS-CoV-2 infection [21,24,147,160,164]. While initially focusing on persistent respiratory system symptoms such as post-infectious asthma and pulmonary fibrosis, symptoms soon expanded to the cardiovascular, endocrine, and nervous systems [165–169]. Persistent anosmia and dysosmia were among the first reported long symptoms of the nervous system, with fatigue and tiredness, insomnia, mood changes, and difficulties in concentrating soon after that [170–175]. According to some studies, the long-lasting nervous system sequelae of COVID-19 are highly selective, with individuals reporting cognitive deficit and depression while alertness and orientation remained intact [174]. These symptoms are more likely to develop in patients with severe clinical disease but have also been described as the only symptom of the infection [24,99,170,175]. While the mechanisms for this remain widely unknown, functional studies report that most patients experiencing these symptoms show biomarkers of protracted inflammation, while cortical areas of the brain are hypometabolic, especially in the frontal and parietal lobes [173]. Thankfully these memory-related areas of the brain show a gradual

improvement in their metabolic function, with most patients slowly regaining baseline cognition functions [173].

*3.6. COVEP*

Based on the accumulated body of evidence, despite some facts being disputable, SARS-CoV-2 infection is a true multisystem disease that affects the central nervous system [10,17,24,39,69,77–79,106,111,115,131,135,162,176,177]. Despite some evidence pointing toward active viral replication within neurons and others disputing it, the neuropathological consensus, for the most part, is that during active COVID-19 infection, there is a presence of neuroinflammation, neuronal degeneration, and an admixture of vascular phenomena such as endotheliitis, blood–brain barrier disruption, and vascular incidents, as well as hypoxic changes (Figure 2) [22,71,79,101,131,176]. Despite some of the presented evidence being contradictory to the others, the protracted central nervous system complaints reported from patients surviving SARS-CoV-2 infection show disruption of brain metabolism and function, with a steady improvement in overall cognitive functioning [173]. Hence, a proposed nomenclature change would be to implement the terminological use of COVID-associated encephalopathy (COVEP). The proposed nomenclature can unite the neuroinflammatory nature with vascular and degenerative phenomena observed in the brains of the deceased as well as the neurocognitive function of survivors, further underlining the truly multifaceted nature of COVID-19 and the multiple systems it can affect with varying clinical severity other than the respiratory system, as with similar nomenclature changes that have already been implemented, e.g., COVAN [10].

## COVEP

*SARS-CoV-2 infection*

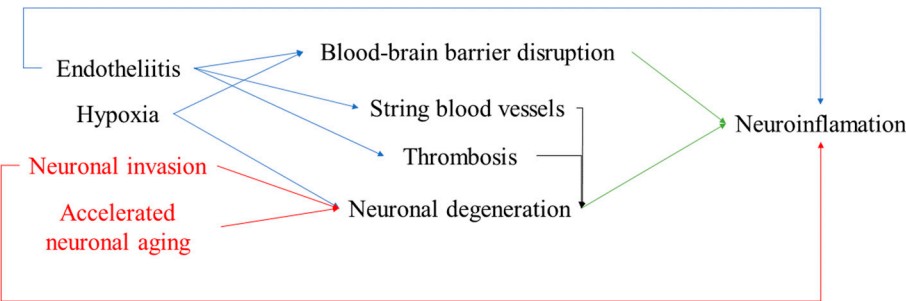

**Figure 2.** Mechanisms of neurological damage associated with COVID-19—COVEP; red denotes disputed mechanisms.

*3.7. Lessons from the Past—Previous Pandemics and von Economo Encephalitis*

COVID-19 is not the first coronavirus disease to cause a severe health concern, nor is it likely to be the last [30]. In the previous two decades, closely related members of the betacoronavirus subfamily—SARS and MERS—have also caused outbreaks of worrying diseases [1,6,29,30,178]. SARS-CoV and MERS-CoV are also highly unlikely to be the first coronaviruses to cause severe illness in humans, as coronaviruses are known to cause severe disease in various animals and have accompanied humanity for most of its recent existence [36,129,179–183]. Despite serological evidence lacking, based on the clinical picture and symptoms reported by the patients, some of which are highly similar to those induced by SARS-CoV-2 (mainly loss of taste and smell, clouding of consciousness, and encephalitis symptoms, all rarely seen in influenza), it is highly likely that the last great pandemic of the nineteenth century (1889–1891), the so-called Russian or Asian influenza, was also caused by a member of the coronavirus family [184,185].

Furthermore, another alarm bell rung based on the clinical manifestations of COVEP and a previous epidemic central nervous system disease was that of the probability of COVEP developing into a von Economo or von Economo-like lethargic encephalitis [100]. While von Economo disease is frequently misinterpreted as post-infectious parkinsonism developing in patients after the 1918–1920 Spanish influenza pandemic, it is essential to note that the first cases were described in 1917, well over a year before the onset of Spanish influenza [100,186–188]. Furthermore, since then serological data from sporadic cases do not confirm influenza viral particles, while being more indicative of an enterovirus causative agent [189]. Multiple reports have since tried to bridge the gap between COVID-19 and von Economo encephalitis, however, the main presenting symptoms of von Economo's of somnolent-ophthalmoplegic, hyperkinetic, and amyostatic-akinetic forms with the development of post-encephalic parkinsonism do not fit the spectrum of clinical sequelae of COVEP [187].

## 4. Conclusions

- COVID-associated encephalopathy—COVEP—is a common clinical manifestation in severe cases of SARS-CoV-2 infection.
- The basis of COVEP is the presence of neuroinflammatory changes, predominantly located in the brainstem and temporal lobe; vascular, degenerative, and hypoxic phenomena also play a significant role in its development.
- These combined changes result in altered sensation, e.g., smell and taste; mood swings; cognitive impediments; insomnia; and severe complications—infarctions, hemorrhages, and neurodegeneration, with these symptoms persisting some time after the infection has been eradicated.
- The neurotropism, neurovirulence, and neuroinvasion of SARS-CoV-2 are still disputable in their mechanisms and direct and indirect effects on the central nervous system.

**Author Contributions:** G.S.S. and A.B.T. conceptualized the manuscript; G.S.S., H.P. and L.P. performed detailed review on clinical and pathological phenomena; M.I. and L.P. performed detailed review on neuroanatomy and processes; G.S.S., H.P. and L.P. wrote the initial draft; M.I. conceptualized the figures; H.P., L.P., M.I. and D.S. performed major text edits; A.B.T. critically reviewer the manuscript and approved its version submitted for publication. All authors have read and agreed to the published version of the manuscript.

**Funding:** This research received no external funding.

**Institutional Review Board Statement:** Not applicable.

**Data Availability Statement:** Not applicable.

**Conflicts of Interest:** The authors declare no conflict of interest.

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
