# Peer review of "COVID-19-Associated Encephalopathy (COVEP): Basic Aspects of Neuropathology"

_encyclopedia, doi:10.3390/encyclopedia2040122_

Round 1
Reviewer 1 Report
This review by Stoyanov et al. describes COVID-19 associated encephalopathy by describing neuroinvasion, regional effects and damage, cellular responses and impacts on nervous system function. Overall, this is a well-written review that cites relevant literature and would benefit those in this field as well as general audiences. There are no substantial weaknesses from the perspective of this reviewer. I have two minor changes that I recommend for the authors:
1) When discussing neuroinvasion, the authors should highlight that monocytes and macrophages can also be infected and therefore could potentially play a role in this process.
2) It would also be beneficial for the authors to include a diagram or image to show how neuroinvasion can occur for SARS-CoV-2.
Author Response
This review by Stoyanov et al. describes COVID-19 associated encephalopathy by describing neuroinvasion, regional effects and damage, cellular responses and impacts on nervous system function. Overall, this is a well-written review that cites relevant literature and would benefit those in this field as well as general audiences. There are no substantial weaknesses from the perspective of this reviewer. I have two minor changes that I recommend for the authors:
- When discussing neuroinvasion, the authors should highlight that monocytes and macrophages can also be infected and therefore could potentially play a role in this process.
- Thank you for these kind comments to our entry manuscript! As you will see we have added the mechanism in the section describing the pathways of entry.
- It would also be beneficial for the authors to include a diagram or image to show how neuroinvasion can occur for SARS-CoV-2.
- As neuroinvasion is a disputed mechanism, as well as the second reviewer suggesting additional changes to the terminology with the introduction of neuroinvasion, neurovirulence and neuroinvasion, we would kindly like to omit the introduction of this change.
Reviewer 2 Report
The authors, G.S. & D. Stoyanov, M. Ivanov, A.B. Tonchev, H. Popov and L. Petkova, in their contribution “COVID-19-associated encephalopathy (COVEP): basic aspects of neuropathology” review and discuss the current understanding of the wide range of neurological complications observed the acute and postacute stages of the COVID-19. An emphasis is placed specifically on the distinctive characteristics that underlie the pathophysiology of the encephalopathy linked to SARS-CoV-2.
The review is, in my opinion, relevant, in-depth, and written with a wide audience in mind. It has a solid structure, extensive research, and a compelling flow. It appears to live up to the expectations raised by its title and is unquestionably interesting to a wide audience.
However, there are several minor issues that the writers need to address.
1) The virus's neuroinvasiveness is an important topic that is covered in detail throughout the publication. However, the authors throughout the text do not use two terms that are undoubtedly clarifying when studying the pathogenesis of the infection: neurotropism and neurovirulence. I would advise the authors to introduce both terms in their discourse in the appropriate sections. Readers would benefit from such a definition.
The following quotation from Bauer et al. (TINS, 2022) is particularly helpful in this regard and should definitely be included.
Bauer, L., Laksono, B. M., de Vrij, F., Kushner, S. A., Harschnitz, O., & van Riel, D. (2022). The neuroinvasiveness, neurotropism, and neurovirulence of SARS-CoV-2. Trends in neurosciences, 45(5), 358–368. https://doi.org/10.1016/j.tins.2022.02.006
2) In the key subsection “Olfactory epithelium and olfactory bulbs”, mores specifically in Line 253: “Initial histopathology suggested changes associated with direct viral-induced damage to the bulbs with edema, inflammatory cell infiltration, microglial nodules and severe neuronal degeneration, neuronal necrosis with areas of colliquation, suggestive of necrotizing olfactory bulbitis (Figure 1). Further reports supported the presence of neuroinflammation and degenerative changes within the olfactory bulbs and added endotheliitis-associated blood-brain barrier injury and suggested the presence of viral RNA.”
It is missing a reference to the important contribution of the group of Peter Mombaerts: Kahn et al. (Cell, 2021) in which it is clearly demonstrated, in an impeccable, revealing and state of the art experimental design, not only the neuropathology of the infection in the mucosa of the nasal cavity, but also how the viral particles do not reach the olfactory bulb. Not only are there no viral particles in the olfactory bulbs of patients who died as a result of Covid, from whom samples were taken just after death, but the olfactory nerves and the olfactory bulb itself remain unaffected.
Khan, M., Yoo, S. J., Clijsters, M., Backaert, W., Vanstapel, A., Speleman, K., Lietaer, C., Choi, S., Hether, T. D., Marcelis, L., Nam, A., Pan, L., Reeves, J. W., Van Bulck, P., Zhou, H., Bourgeois, M., Debaveye, Y., De Munter, P., Gunst, J., Jorissen, M., … Van Gerven, L. (2021). Visualizing in deceased COVID-19 patients how SARS-CoV-2 attacks the respiratory and olfactory mucosae but spares the olfactory bulb. Cell, 184(24), 5932–5949.e15. https://doi.org/10.1016/j.cell.2021.10.027
3) Line 259: “A promising sign for the reversible nature of the SARS-CoV-2 induced nervous system pathology is the recovery of smell sensation, albeit delayed in most patients after COVID-19, probably linked to adult neurogenesis in the bulb.”
Given the minimal alterations observed in the olfactory bulb, the regenerative capability of the olfactory receptor neurons and their axonal processes seems more relevant for olfactory recovery. This enables the regeneration of the olfactory cells that have been lost as a result of the infection of their sustentacular cells population. Furthermore, the role of adult neurogenesis in the olfactory bulb seems very limited from the point of view of combating degenerative diseases affecting the bulb.
In all this context this reference is very pertinent:
Baig AM. Loss of smell in COVID-19: reasons for variable recovery patterns from anosmia. Neural Regen Res. 2022 Jul;17(7):1623-1624. doi: 10.4103/1673-5374.330625. PMID: 34916450; PMCID: PMC8771103.
4) Are there differences among SARS-CoV-2 variants, in particular variants of concern, in their neuroinvasiveness and in the COVEP incidence?
This issue should be addressed.
5) Although it is discussed throughout the paper in the case of Parkinson's disease, the question arises as to whether infection with SARS-CoV-2 can exacerbate other neurodegenerative or neuropsychiatric diseases, such as Alzheimer's, multiple sclerosis, schizophrenia, depression, etc. Could the authors elaborate on this?
6) Minor issue:
Line 350: “do not fir the spectrum” should be fit.

Author Response
The authors, G.S. & D. Stoyanov, M. Ivanov, A.B. Tonchev, H. Popov and L. Petkova, in their contribution “COVID-19-associated encephalopathy (COVEP): basic aspects of neuropathology” review and discuss the current understanding of the wide range of neurological complications observed the acute and postacute stages of the COVID-19. An emphasis is placed specifically on the distinctive characteristics that underlie the pathophysiology of the encephalopathy linked to SARS-CoV-2.
- Thank you for these kind comment towards our entry manuscript!
The review is, in my opinion, relevant, in-depth, and written with a wide audience in mind. It has a solid structure, extensive research, and a compelling flow. It appears to live up to the expectations raised by its title and is unquestionably interesting to a wide audience.
- Thank you!
However, there are several minor issues that the writers need to address.
- As you will see, we have implemented the suggested changes throughout the relevant areas of the manuscript.
1) The virus's neuroinvasiveness is an important topic that is covered in detail throughout the publication. However, the authors throughout the text do not use two terms that are undoubtedly clarifying when studying the pathogenesis of the infection: neurotropism and neurovirulence. I would advise the authors to introduce both terms in their discourse in the appropriate sections. Readers would benefit from such a definition.
The following quotation from Bauer et al. (TINS, 2022) is particularly helpful in this regard and should definitely be included.
Bauer, L., Laksono, B. M., de Vrij, F., Kushner, S. A., Harschnitz, O., & van Riel, D. (2022). The neuroinvasiveness, neurotropism, and neurovirulence of SARS-CoV-2. Trends in neurosciences, 45(5), 358–368. https://doi.org/10.1016/j.tins.2022.02.006
- Thank you for this key suggestion, regarding he mechanisms for the development of encephalopathy! We have introduced and discussed the terminology in the initial section of the manuscript.
2) In the key subsection “Olfactory epithelium and olfactory bulbs”, mores specifically in Line 253: “Initial histopathology suggested changes associated with direct viral-induced damage to the bulbs with edema, inflammatory cell infiltration, microglial nodules and severe neuronal degeneration, neuronal necrosis with areas of colliquation, suggestive of necrotizing olfactory bulbitis (Figure 1). Further reports supported the presence of neuroinflammation and degenerative changes within the olfactory bulbs and added endotheliitis-associated blood-brain barrier injury and suggested the presence of viral RNA.”
It is missing a reference to the important contribution of the group of Peter Mombaerts: Kahn et al. (Cell, 2021) in which it is clearly demonstrated, in an impeccable, revealing and state of the art experimental design, not only the neuropathology of the infection in the mucosa of the nasal cavity, but also how the viral particles do not reach the olfactory bulb. Not only are there no viral particles in the olfactory bulbs of patients who died as a result of Covid, from whom samples were taken just after death, but the olfactory nerves and the olfactory bulb itself remain unaffected.
Khan, M., Yoo, S. J., Clijsters, M., Backaert, W., Vanstapel, A., Speleman, K., Lietaer, C., Choi, S., Hether, T. D., Marcelis, L., Nam, A., Pan, L., Reeves, J. W., Van Bulck, P., Zhou, H., Bourgeois, M., Debaveye, Y., De Munter, P., Gunst, J., Jorissen, M., … Van Gerven, L. (2021). Visualizing in deceased COVID-19 patients how SARS-CoV-2 attacks the respiratory and olfactory mucosae but spares the olfactory bulb. Cell, 184(24), 5932–5949.e15. https://doi.org/10.1016/j.cell.2021.10.027
- Thank you for suggesting the reference in this section of the manuscript as well.
3) Line 259: “A promising sign for the reversible nature of the SARS-CoV-2 induced nervous system pathology is the recovery of smell sensation, albeit delayed in most patients after COVID-19, probably linked to adult neurogenesis in the bulb.”
Given the minimal alterations observed in the olfactory bulb, the regenerative capability of the olfactory receptor neurons and their axonal processes seems more relevant for olfactory recovery. This enables the regeneration of the olfactory cells that have been lost as a result of the infection of their sustentacular cells population. Furthermore, the role of adult neurogenesis in the olfactory bulb seems very limited from the point of view of combating degenerative diseases affecting the bulb.
In all this context this reference is very pertinent:
Baig AM. Loss of smell in COVID-19: reasons for variable recovery patterns from anosmia. Neural Regen Res. 2022 Jul;17(7):1623-1624. doi: 10.4103/1673-5374.330625. PMID: 34916450; PMCID: PMC8771103.
- Thank you for suggesting the inclusion of this reference, we have expanded the section on the restoration of smell sensation and cited the relevant article.
4) Are there differences among SARS-CoV-2 variants, in particular variants of concern, in their neuroinvasiveness and in the COVEP incidence?
This issue should be addressed.
- Sadly, there is limited literature data on this question. Undoubtedly the less pathogenic the variant is, the lower the likelihood of neuropathology developing, however that assumption does not always translate into practice, with the probability of a less pathogenic variant to the respiratory system also developing more of a neurovirulent trait as well.
5) Although it is discussed throughout the paper in the case of Parkinson's disease, the question arises as to whether infection with SARS-CoV-2 can exacerbate other neurodegenerative or neuropsychiatric diseases, such as Alzheimer's, multiple sclerosis, schizophrenia, depression, etc. Could the authors elaborate on this?
- As mentioned in the text and not only for neurological conditions, but the discontinuation of systemic therapy has also led to deterioration in most of these patients. Secondly the only available data is on the spectrum of neuropsychiatric conditions, where not so much the effects of the virus, but again the social consequence have impacted in a bad way. There is real world data on exacerbation of Parkinson’s patients, however we dare not expand this section further, as to avoid it being misinterpreted as another sigh for encephalitis lethargica association, as this condition has different morphology and has been widely misinterpreted in the context of COVID, both in its etiology, manifestation and sequele.
6) Minor issue:
Line 350: “do not fir the spectrum” should be fit.
- Thank you for pointing out this typographical error!
Round 2
Reviewer 1 Report
The authors adequately addressed my comments.
Author Response
The authors adequately addressed my comments.
- Thank you for this kind comment regarding the edits applied to the manuscript.
Reviewer 2 Report
The authors have satisfactorily addressed my concerns and revised the manuscript accordingly. I would recommend the manuscript for publication in Encyclopedia.
Author Response
The authors have satisfactorily addressed my concerns and revised the manuscript accordingly. I would recommend the manuscript for publication in Encyclopedia.
- Thank you for these kind comment towards our entry manuscript!